

# SDUST2023BCO: a global seafloor model determined from multi-layer perceptron neural network using multi-source differential marine geodetic data

Shuai Zhou [1, 2], Jinyun Guo [1*], Huiying Zhang [1], Yongjun Jia [3], Heping Sun [2], Xin Liu [1], Dechao An [4]

[1]College of Geodesy and Geomatics, Shandong University of Science and Technology, Qingdao 266590, China
[2]State Key Laboratory of Geodesy and Earth's Dynamics, Innovation Academy of Precision Measurement Science and Technology, Chinese Academy of Sciences, Wuhan 430077, China
[3]National Satellite Ocean Application Service, Ministry of Natural Resources, Beijing 100812, China
[4]School of Geospatial Engineering and Science, Sun Yat-sen University, Zhuhai 519082, China
* *Correspondence to*: Jinyun Guo (jinyunguo1@126.com)

**Abstract.** Seafloor topography, as a fundamental marine spatial geographic information, plays a vital role in marine observation and science research. With the growing demand for high-precision bathymetric models, the Multi-layer Perceptron (MLP) neural network is used to integrate multi-source marine geodetic data in this paper. A new bathymetric model of the global ocean, spanning 0°-360°E and 80°S-80°N, has been constructed, known as the Shandong University of Science and Technology 2023 Bathymetric Chart of the Oceans (SDUST2023BCO), with a grid size of 1′×1′. The multi-source differential marine geodetic data used include gravity anomaly data released by Shandong University of Science and Technology, vertical gravity gradient and the vertical deflection data released by Scripps Institution of Oceanography, as well as mean dynamic topography data released by the Centre National d'Etudes Spatiales. First, input and output data are organized from the multi-source marine geodetic data to train the MLP model. Second, the input data at interesting points are fed into the MLP model to obtain prediction bathymetry at interesting points. Finally, a high-precision bathymetric model with a resolution of 1′×1′ has been constructed for the global marine area. The accuracy of the bathymetric model is evaluated by comparing with single-beam shipborne bathymetric data, and GEBCO_2023 and topo_25.1 models. The results demonstrate that the SDUST2023BCO model is accurate and reliable, effectively capturing and reflecting global ocean bathymetric information. The SDUST2023BCO model is available at https://doi.org/10.5281/zenodo.13341896 (Zhou et al., 2024).

## 1 Introduction

As a critical foundational dataset for marine scientific research, global bathymetric information plays a vital role in multiple disciplines such as marine geodesy, geophysics, biology and seafloor geology. It is also essential for marine economic development, oceanographic surveys, maritime navigation and rescue operations (Hirt and Rexer, 2015; Hu et al., 2015; Yang et al., 2018; Sandwell et al., 2022). Currently, shipborne single-beam bathymetric techniques can provide high-precision bathymetric data, which is one of the most direct ways for detecting seafloor topography. However, despite the



accumulation of data collected through shipborne techniques, large areas of the global oceans, especially in the Southern Hemisphere, remain largely uncharted (Hu et al., 2014). Moreover, shipborne single-beam bathymetric data, characterized

by its low resolution, expensive expenses and low precision in positioning and measurements of older datasets, presents significant limitations (Hu et al., 2014; Xing et al., 2020). The progression in satellite altimetry technology has ushered in a novel era for the development of bathymetric models. Satellite altimetry, as one of the critical techniques for acquiring global marine data, can obtain the global-coverage, uniformly distributed, high-precision, and high-resolution sea surface heights. The global marine gravity field information can be recovered based on relevant geodetic methods (Marks and Smith,

2012; Sun et al., 2021; Kim et al., 2011). The global bathymetric model can be obtained with seafloor-inversion methods, considering the inherent correlation between seafloor topography and global marine gravity information (Wang, 2000; Hu et al., 2021, Yeu et al., 2018).

Currently, the inversion of bathymetric values based on marine gravity data acquired from satellite altimeter data has become a reliable approach to construct global bathymetric models. The methods employed for predicting seafloor topography based

on satellite altimeter data mainly include frequency-domain methods, spatial-domain methods (analytical methods), least squares collocation methods, and gravity-geological methods (GGM). While these methods have effectively constructed high-precision bathymetric models for specific regions, such as the South China Sea (Fan et al., 2020; An et al., 2022, Hu et al., 2020), the western Pacific Ocean (Yang et al., 2018), the Gulf of Guinea (Annan and Wan, 2020), the Philippine Sea (An et al., 2023) and the New Zealand (Ramillien & Wright, 2000), the nonlinear relationship between gravity data and seafloor

topography is still not adequately used by these methods. At the same time, the seafloor topography is constructed solely based on the linear relationship between gravity anomalies or vertical gravity gradients and the seafloor topography. Consequently, a global bathymetric model can be constructed by integrating the nonlinear components inherent in the relationship between multi-source marine geodetic data and the seafloor topography while accounting for long-wavelength information present in these datasets.

With the continuous advancement in computer storage and computational capabilities, the machine learning or deep learning has been widely applied in various scientific fields, such as environmental science (Sunil et al., 2024), geology (Kuster and Toksoz, 1974), and clinical medicine (Lee et al., 2019). Currently, the machine learning or deep learning methods are increasingly used to construct bathymetric models. Annan & Wan (2022) used Convolutional Neural Networks (CNN) to establish a high-precision bathymetric model of the Gulf of Guinea from multi-source gravity data. However, this method

may introduce bias when processing input data with convolutional layers. Sun et al. (2021) proposed a method combining neural networks and wavelet decomposition of gravity information, and the superiority of this method was validated. However, this model only used gravity anomaly and vertical gravity gradient data, without considering other multi-source marine geodetic data. Zhou et al. (2023) used Multi-Layer Perceptron (MLP) neural network with a regional inversion approach to construct a high-precision bathymetric model of the Gulf of Mexico. However, the impact of long-wavelength

information from multi-source marine geodetic data on the accuracy of the constructed bathymetric model was not considered.

The focus of this paper is the establishment of a new global (0°-360°E, 80°S-80°N) bathymetric model, named Shandong University of Science and Technology 2023 Bathymetric Chart of the Oceans (SDUST2023BCO). This model is constructed based on a MLP neural network, integrating the differences from multi-source marine geodetic data (gravity anomalies, vertical gravity gradients, the meridional and prime components of vertical deflection, mean dynamic topography). The reliability of this model is validated by comparing it with the GEBCO_2023 and topo_25.1 models. Section 2 introduces the multi-source marine geodetic data used in this study. Section 3 explains the processing methods for shipborne single-beam bathymetric data, the principle of MLP neural network, the organization of input/output data, and the procedure for constructing the bathymetric model. Section 4 is the results and discussions and validates the SDUST2023BCO model by comparing it with single-beam shipborne bathymetric data, and GEBCO_2023 and topo_25.1 models. Section 5 is the conclusion.

## 2 Data

The global marine (0°-360°E, 80°S-80°N) is designated as the study region in this paper. Due to the limitations in computational power and storage capacity, the study region is divided into 144 sub-regions, as shown in Fig. 1. From west to east, the area is divided into 18 columns and marked from LON1 to LON18; From north to south, the region is divided into 8 rows and marked from LAT1 to LAT8. To mitigate edge effects and stitching issues between different sub-regions, each sub-region is expanded by 0.1° in all directions. The extended data is used for the inversion of seafloor topography.

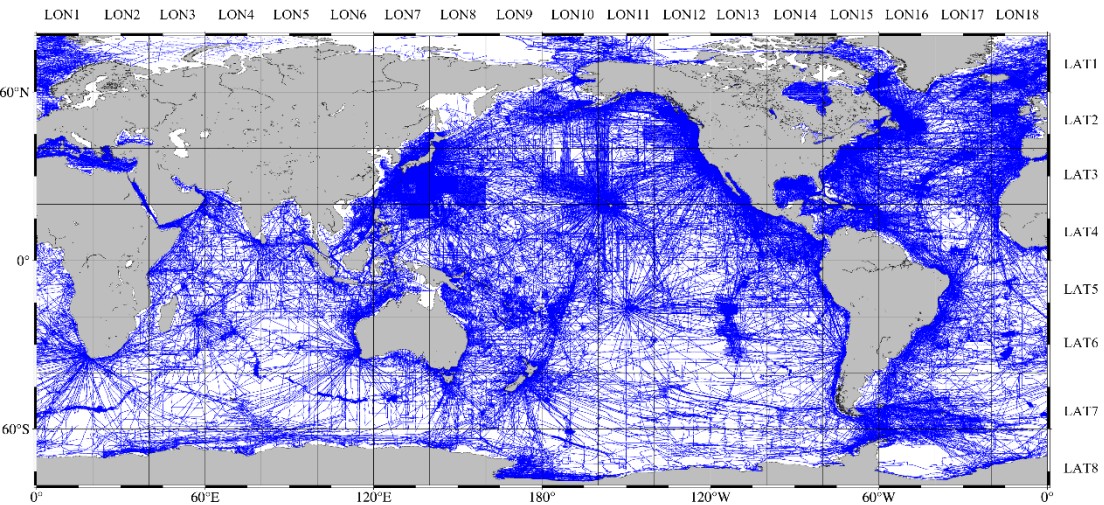

**Figure 1. Division map and the distribution of shipborne single-beam bathymetry data.**



## 2.1 Shipborne single-beam bathymetry data

The shipborne single-beam bathymetry data are provided by the National Centers for Environmental Information (NCEI), a division of the National Oceanic and Atmospheric Administration (NOAA) in the United States. The dataset contains global marine bathymetric data collected since the 1950s. The study area includes 5,374 shipborne single-beam bathymetry tracks, as shown in Fig. 1.

Owing to the large time span of the shipborne single-beam bathymetric data, some datasets have imprecise localization and significant measurement inaccuracies. Therefore, it is necessary to preprocess the data to remove some points with substantial errors. Now, the global marine bathymetric models derived from satellite altimetry data achieve a high level of accuracy. The topo_25.1 model, as the latest bathymetric model released by the Scripps Institution of Oceanography (SIO), shows a standard deviation (STD) of approximately 435m compared to global shipborne single-beam bathymetric data.

Therefore, this paper uses the topo_25.1 model as a prior model to remove shipborne single-beam bathymetric points with significant errors. The process of elimination primarily consists of two parts:

The first step is to remove shipborne single-beam bathymetric tracks that contain significant errors. First, the topo_25.1 model is used to calculate the predicted bathymetry at each shipborne single-beam bathymetric point using a cubic spline interpolation method. The difference between the topo_25.1 predicted bathymetric values and the actual measured bathymetry at each point is calculated, and the standard deviation (STD1) of these differences is calculated. Second, the topo_25.1 model is interpolated onto each shipborne single-beam bathymetric track to obtain corresponding bathymetry. The differences between these interpolated values and the actual measured depths along each track are calculated, and the STD of these differences is computed for each track. Finally, the entire shipborne single-beam bathymetric track is removed if its STD exceeds three times the STD1. Using this method, 38 ship tracks with significantly errors are eliminated, leaving 5,336 shipborne single-beam bathymetric tracks, which consist of a total of 11,335,376 shipborne single-beam bathymetric points.

The second step is to remove shipborne single-beam bathymetric points with large errors. Despite the initial removal of entire tracks, some individual shipborne single-beam bathymetric points with significant large errors may still remain. Therefore, the method is employed to eliminate shipborne single-beam bathymetric points that exhibit significant errors. First, the topo_25.1 model is interpolated onto all the remaining shipborne single-beam bathymetric points to obtain the topo_25.1 predicted bathymetry at these points. The difference between the topo_25.1 predicted bathymetry and the actual measured bathymetry at each point is calculated. The STD of these differences is calculated and shipborne single-beam bathymetric points with absolute bathymetric residuals greater than three times the STD are removed. Finally, the 1,016,374 shipborne single-beam bathymetric points are eliminated, leaving 112,319,002 shipborne single-beam bathymetric points, with a removal rate of 0.90%.

The residual shipborne single-beam bathymetric points are used to train the MLP model which is employed to construct the SDUST2023BCO model.



## 2.2 Marine Geodetic Data

### 2.2.1 Marine Gravity Data

Gravity anomaly data originates from the global gravity anomaly model (SDUST2022GRA) constructed by Shandong
University of Science and Technology in 2022. This model is constructed based on the along-track radar altimeter data (Li et al., 2023) and the accuracy and reliability of this model have been verified by comparising with the DTU17 model, SIO grav_32.1 model, and shipborne gravity data from NCEI. The precision of the SDUST2022GRA model in local coastal and high-latitude areas has been effectively enhanced. The model is available for download at https://doi.org/10.5281/zenodo.8337387, with a resolution of 1'×1'.

Based on the correlation between vertical deflection, vertical gravity gradient, and bathymetry, those data can also be utilized to predict bathymetry. These gravity data are derived from the 32.1 version released by SIO in 2022, with a resolution of 1'×1', and can be freely obtained from https://topex.ucsd.edu/pub/global_grav_1min/. This version of the gravity models, which determines from Cryosat LRM, Altika, Cryosat-SAR, and Sentinel-3A/B, has high quality and precision.

### 2.2.2 MDT model

The MDT model used in this study is the MDT-CNES-CLS18 model released by the Centre National d'Etudes Spatiales. This model plays a crucial role in land-sea elevation data, physical oceanography, and global climate change studies (Woodworth et al., 2015), and it can be downloaded from https://www.aviso.altimetry.fr/en/data/products/. This model has a resolution of 7.5'×7.5', with foundational data primarily sourced from the CNES-CLS15 mean sea surface model (Pujol et al., 2018), the GOCO05S geoid model, hydrographic data, and drifter data.

### 2.2.3 Bathymetric Models


To validate the accuracy of the SDUST2023BCO model, this paper introduces the GEBCO_2023 model and the topo_25.1 model.

The GEBCO_2023 model, released in 2023 by the Nippon Foundation-GEBCO Seabed 2030 Project, is a global elevation model developed in collaboration between the Nippon Foundation (Japan) and GEBCO (General Bathymetric Chart of the
Oceans). It covers the latitude range from 90°N to 90°S with a resolution of 15", and can be downloaded from https://www.gebco.net.

The topo_25.1 model, released by the SIO in 2023, is the 25.1 version of the global bathymetric model. It covers latitudes from 80°N to 80°S with a resolution of 1'×1'. The model is available at https://topex.ucsd.edu/pub/global_topo_1min/.



## 3 Methodology

### 3.1 MLP Neural Network

Neural networks, which do not rely on explicit mathematical expressions between inputs and outputs, can learn and model nonlinear relationships between input and output vectors, facilitating complex function approximation. They have a strong capacity to learn the intrinsic features of datasets, which has been applied in numerous domains (Jin et al., 2021; Kuremoto et al., 2014).

The MLP neural network, as a machine learning method, is a type of feedforward neural network. An MLP neural network consists of an input layer, an output layer, and a number of hidden layers, which can be adjusted according to the practical requirements. Each layer is composed of several neurons, also known as nodes. Layers within the network are fully connected, meaning that each neuron in one layer is linked to every neuron in the next layer. Due to the linear connections between neurons across different layers, activation functions are introduced to enhance the nonlinearity. Consequently, the output of a neuron can be expressed as:

$$y = f(Wx + b) \tag{1}$$

where $x$ and $y$ represent the input and the output data from a neuron, $W$ and $b$ represent the weights and biases, $f(\bullet)$ represents the tanh activation function in this paper, which adds the nonlinearity to the MLP neural network, enabling it to approximate complex functions. The activation function allows the model to learn and fit complex patterns in the dataset.

### 3.2 Organization of Input/Output Data

The organization format of input/output data significantly influences the training and predictive precision of MLP neural networks. Based on the correlation between gravity data and bathymetry (Smith & Sandwell, 1994), the gravity anomaly, the meridional and prime components of vertical deflection are used as input data for training and prediction. Since MDT data can reflect the variations in bathymetric models, MDT data has also been introduced.

The bathymetry at a particular point is influenced by various factors in its surroundings, the more surrounding points there are, the more information is provided (Zhu et al., 2021, 2023). Due to the limitations in computational processing power and memory storage, an 8' × 8' grid centered on each interesting point is constructed by extending outward from each point. Grid nodes on the 8'×8' grid are marked from node 1 to node 64, as shown in Fig. 2. To mitigate the impact of long-wavelength information in multi-source geodetic data, this paper uses the differences between the multi-source marine geodetic data at each grid point within an 8'×8' area surrounding the interesting point and the multi-source marine geodetic data at the interesting point. These differences are employed as the input data to train the MLP neural network.



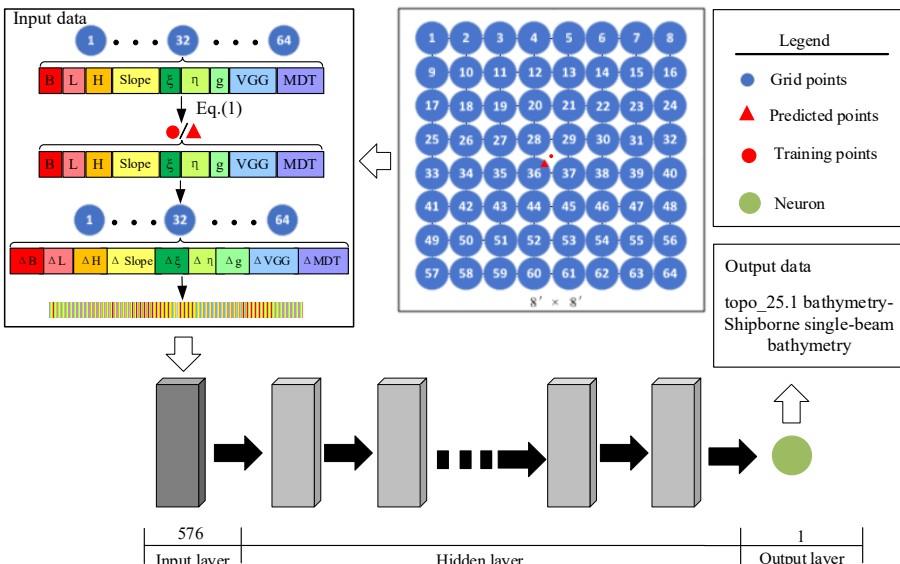

**Figure 2. The organization of input/output data and structure of MLP.**

The training dataset includes all shipborne single-beam bathymetric point within global marine, which will be referred to as the training points. The input data used for training in this paper are the differences of multi-source marine geodetic data, which include location information (longitude, latitude), bathymetry, slope, the meridional components of vertical deflection, the prime components of vertical deflection, vertical gravity gradient, gravity anomaly and MDT data. The relevant calculation equation is as follows:

$$\begin{cases} \Delta L = L^i_{grid} - L_s \\ \Delta B = B^i_{grid} - B_s \\ \Delta h = h^i_{grid} - h_s \\ \Delta slope = slope^i_{grid} - slope_s \\ \Delta\xi = \xi^i_{grid} - \xi_s \qquad\qquad (i=1,2...64) \\ \Delta\eta = \eta^i_{grid} - \eta_s \\ \Delta(\Delta g) = \Delta g^i_{grid} - \Delta g_s \\ \Delta VGG = VGG^i_{grid} - VGG_s \\ \Delta MDT = MDT^i_{grid} - MDT_s \end{cases} \qquad (2)$$

where i represents the i-th grid node; $L^i_{grid}$, $B^i_{grid}$ are the longitude and latitude at the i-th grid node, while $L_s$, $B_s$ represent the longitude and latitude at the training or prediction point; $h^i_{grid}$, $\xi^i_{grid}$, $\eta^i_{grid}$, $\Delta g^i_{grid}$, $VGG^i_{grid}$, $MDT^i_{grid}$ represent the





interpolations of the topo_25.1 model, the SIO meridional components of vertical deflection model, the SIO prime components of vertical deflection model, the SDUST2022GRA gravity anomaly model, the SIO vertical gravity gradient model, and the MDT model at the i-th grid node; $h_s$, $\xi_s$, $\eta_s$, $\Delta g_s$, $VGG_s$ and $MDT_s$ represent the interpolations of the topo_25.1 model, the SIO meridional components of vertical deflection model, SIO prime components of vertical deflection

model, the SDUST2022GRA gravity anomaly model, the SIO vertical gravity gradient model, and the MDT model at the training or prediction point. The slope, defined as the ratio of the difference in seafloor height to distance, is calculated by the following equation:

$$slope_i = \frac{h_i - h_{i+1}}{\sqrt{(x_i - x_{i+1})^2 + (y_i - y_{i+1})^2}} \qquad (3)$$

where $slope_i$ represents the slope of the target point at the i-th location, $h_i$ is the bathymetry at the i-th point, and $h_{i+1}$ is the bathymetry at the i+1-th point, which is 1' longitudinally or latitudinally apart from the i-th point. $x_i$, $y_i$ are the horizontal

and vertical coordinates of the i-th point, while $x_{i+1}$, $y_{i+1}$ are the corresponding coordinates of the i+1-th point. Using Eq. (3), the slopes in four directions -longitudinal and latitudinal- are calculated. The maximum value among these four directional slopes is taken as the final slope for the target point.

The output data used for training is:

$$h_{output}^{train} = h_s - h \qquad (4)$$

**3.3 Method for model construction**

The procedure for constructing a global marine bathymetric model using an MLP neural network is illustrated in Table 1, with the specific steps detailed as follows:

The first step is the standardization of the input and output data. Due to the significant differences in magnitude between various types of data, the input and output data must be standardized to eliminate the effects of dimensional discrepancies. The calculation equation is as follows:

$$\hat{x} = \frac{x_i - \overline{x}}{\sigma} \qquad (5)$$


where $\hat{x}$ represents the standardized data, $x_i$ is the data before standardization, $\overline{x}$ is the mean of the input data, and $\sigma$ is the STD of the input data. After standardization, the mean of the input data becomes 0 and the STD becomes 1, ensuring that all input data contribute equally to the training of the MLP neural network.

The second step is to select appropriate neural network parameters. The choice of parameters is critical for the training and

prediction accuracy of the MLP model. This includes the initialization of weights and biases, the number of hidden layers, activation function, learning rate, and batch size. In order to achieve high precision in training and prediction, the selection





of parameters may be adjusted in different sub-regions. This paper chiefly adopts a network design with four hidden layers, with neuron counts of 512, 256, 128, and 64 respectively, a learning rate of 0.0001, and a training batch size of 8.

The third step is the training of the MLP model. First, the MLP neural network is trained using the input and output data.
Second, an appropriate loss function and optimization algorithm should be selected. Finally, the MLP neural network models for 144 sub-regions are established through training. In this paper, the mean squared error (MSE) is chosen as the loss function, and the Adam optimization algorithm is used to update the weights and biases.

The fourth step is the calculation of bathymetric values. Based on step (3), 144 MLP neural network models for the sub-regions are established. The prediction outcomes for each sub-region are obtained by feeding the input data into the
corresponding MLP models for all 144 sub-regions. Since the prediction result is the differences between the topo_25.1 model at the prediction points and the actual bathymetry value at these points, the equation for calculating the predicted bathymetry value is:

$$h_{output}^{pred} = \Delta h'_{result} + h'_{topo\_25.1} \tag{6}$$

where $h_{output}^{pred}$ represents the predicted bathymetry value at the prediction point, $\Delta h'_{result}$ represents the prediction output result of the MLP model, and $h'_{topo\_25.1}$ represents the bathymetry interpolation from the topo_25.1 model at the prediction point.

The final step is the construction of the global bathymetric model. Due to each sub-region has been extended outward by 0.1°, the average bathymetry of the overlapping areas is taken as the final bathymetric value. This method ensures a smooth transition between sub-regions and avoids any abrupt changes in the bathymetric model. By integrating all sub-regions, a new global bathymetric model is constructed.

**Table 1. The algorithm of MLP neural network for constructing the seafloor topography model.**

| **Algorithm 1:** MLP neural network algorithm for constructing the seafloor topography model. |
| --- |
| 1. **Input**: Training set **T** and prediction set **P** for each sub-region. |
| 2. **Initialization**: Normalize the datasets using the Eq. (5); **N** denotes the number of iterations. |
| 3. **for i = 1 to N do** |
| 4.   Initialize the weights **W** and biases **b**. |
| 5.   Compute the output of each neuron in each layer and the final output, using Eq. (1). |
| 6.   Calculate the loss function. |
| 7.   Update the weights **W** and biases **b** using the Adam algorithm. |
| 8. **end for** |
| 9. **until** the maximum number of iterations is reached or the loss function no longer decreases; |
| 10. Save the MLP model. |
| 11. Obtain prediction values using the prediction set P. |
| 12. Recover the bathymetry values using Eq. (6). |
| 13. Bathymetry models for each sub-region. |
| 14. **Output:** Global bathymetry model. |



## 4. Results and Analysis

### 4.1 Training Results of the MLP Neural Network

First, input and output data are organized according the Sect. 3.2. Second, the MLP neural network is trained with those data to establish MLP models for each sub-region. Throughout the training phase, the weights within the MLP neural network are iteratively adjusted via the Adam optimization algorithm. The training outcomes gradually converges to the actual bathymetric values, and the MLP models for each sub-region are constructed. The training accuracy for each sub-region is showed in Table 2. Table 2 shows that the training accuracy for approximately 91.4% of the sub-regions exceeds 95%. This indicates that the MLP models constructed for these sub-regions have achieved a high level of accuracy. This satisfies the requirements for predicting bathymetry, demonstrating the effectiveness of these models for this application.

**Table 2. Training accuracy of each sub_region.**

| Aera | Evaluation Metrics | LAT1 | LAT2 | LAT3 | LAT4 | LAT5 | LAT6 | LAT7 | LAT8 |
|---|---|---|---|---|---|---|---|---|---|
| LON1 | $R^2$/% | 98.32 | 98.61 | 99.32 | 95.04 | 93.69 | 96.15 | 98.23 | 97.98 |
| | STD/m | 10.69 | 8.07 | 10.83 | 25.68 | 14.46 | 5.64 | 9.45 | 3.84 |
| LON2 | $R^2$/% | 98.41 | 96.88 | 99.27 | 98.80 | 98.01 | 99.33 | 98.33 | 98.96 |
| | STD/m | 0.91 | 3.69 | 7.19 | 4.57 | 1.21 | 5.22 | 8.63 | 8.41 |
| LON3 | $R^2$/% | 98.09 | 93.69 | 98.96 | 96.50 | 97.58 | 99.26 | 97.17 | 97.43 |
| | STD/m | 1.77 | 2.51 | 5.25 | 11.18 | 6.99 | 9.34 | 6.99 | 6.52 |
| LON4 | $R^2$/% | 95.94 | - | 97.78 | 97.45 | 97.55 | 98.97 | 96.90 | 97.03 |
| | STD/m | 1.67 | - | 5.28 | 7.10 | 9.56 | 9.62 | 8.00 | 5.75 |
| LON5 | $R^2$/% | 92.55 | - | 99.71 | 98.61 | 96.04 | 98.63 | 97.84 | 97.89 |
| | STD/m | 2.19 | - | 1.55 | 3.55 | 6.53 | 5.11 | 5.98 | 5.21 |
| LON6 | $R^2$/% | 86.68 | - | 95.36 | 97.19 | 96.91 | 96.71 | 98.92 | 96.93 |
| | STD/m | 0.58 | - | 2.33 | 10.52 | 9.56 | 10.32 | 5.90 | 3.76 |
| LON7 | $R^2$/% | 97.23 | 98.23 | 99.06 | 95.77 | 98.18 | 97.66 | 98.57 | 98.88 |
| | STD/m | 3.91 | 6.87 | 8.68 | 12.51 | 7.29 | 6.34 | 7.27 | 5.15 |
| LON8 | $R^2$/% | 98.68 | 97.94 | 93.07 | 93.39 | 98.90 | 96.40 | 97.05 | 97.47 |
| | STD/m | 2.03 | 9.83 | 17.71 | 15.28 | 13.51 | 7.79 | 10.31 | 5.82 |
| LON9 | $R^2$/% | 97.21 | 97.49 | 95.90 | 96.99 | 95.95 | 95.50 | 94.19 | 95.66 |
| | STD/m | 3.74 | 9.99 | 9.57 | 10.62 | 13.91 | 7.20 | 9.09 | 8.62 |
| LON10 | $R^2$/% | 99.27 | 95.99 | 96.93 | 97.15 | 96.19 | 94.44 | 97.48 | 97.97 |
| | STD/m | 4.23 | 11.94 | 13.04 | 11.42 | 15.51 | 12.54 | 6.68 | 7.38 |
| LON11 | $R^2$/% | 98.85 | 97.80 | 96.59 | 97.30 | 97.16 | 96.94 | 97.95 | 98.10 |
| | STD/m | 6.53 | 9.98 | 11.18 | 9.54 | 11.39 | 7.16 | 7.35 | 6.93 |
| LON12 | $R^2$/% | 98.24 | 96.09 | 96.78 | 96.34 | 96.71 | 97.81 | 97.16 | 95.91 |
| | STD/m | 8.31 | 9.05 | 8.23 | 6.61 | 7.57 | 5.85 | 9.29 | 5.21 |
| LON13 | $R^2$/% | 94.12 | - | 94.54 | 97.08 | 96.77 | 98.47 | 98.99 | 96.96 |
| | STD/m | 1.45 | - | 9.52 | 10.85 | 9.54 | 9.21 | 5.83 | 3.14 |





| | | | | | | | | | |
|---|---|---|---|---|---|---|---|---|---|
| LON14 | R$^2$/% | 99.16 | 99.90 | 97.57 | 96.71 | 96.01 | 97.98 | 98.53 | 95.81 |
| | STD/m | 1.95 | 1.77 | 9.70 | 11.02 | 11.51 | 8.39 | 6.93 | 2.59 |
| LON15 | R$^2$/% | 99.87 | 97.10 | 98.57 | 95.25 | 97.18 | 98.71 | 98.09 | 97.50 |
| | STD/m | 3.47 | 4.01 | 5.17 | 14.25 | 7.37 | 9.25 | 8.06 | 5.20 |
| LON16 | R$^2$/% | 97.11 | 96.73 | 97.16 | 95.56 | 99.24 | 98.97 | 95.69 | 98.38 |
| | STD/m | 6.83 | 7.09 | 14.59 | 13.03 | 3.31 | 6.81 | 11.64 | 3.59 |
| LON17 | R$^2$/% | 97.30 | 97.22 | 97.86 | 98.73 | 98.32 | 94.95 | 99.15 | 94.63 |
| | STD/m | 4.88 | 12.31 | 13.59 | 10.07 | 9.17 | 4.29 | 9.85 | 8.41 |
| LON18 | R$^2$/% | 96.26 | 95.97 | 96.74 | 95.90 | 96.16 | 98.23 | 99.62 | 95.21 |
| | STD/m | 9.37 | 10.04 | 11.34 | 9.07 | 13.44 | 11.74 | 7.55 | 6.42 |

Note: "-" indicates that the area is land or does not have shipboard bathymetric soundings.

## 4.2 Bathymetric model based on MLP neural network

Input data at the prediction points within each sub-region is fed into the respective MLP model, the predicted bathymetry for the center points of each 1'×1' grid are obtained. The predicted bathymetry is the difference between the STUST2023BCO model and the topo_25.1 model. Figure 3 presents the difference map between the two models, illustrating that the

discrepancies are mainly centered around 0m. According to statistics, the ratio of differences that fall within the range of ±100m is 96.89%. The high correlation and minimal differences between the two models, as revealed by this analysis, further validate the effectiveness of the MLP neural network method in constructing bathymetric models.

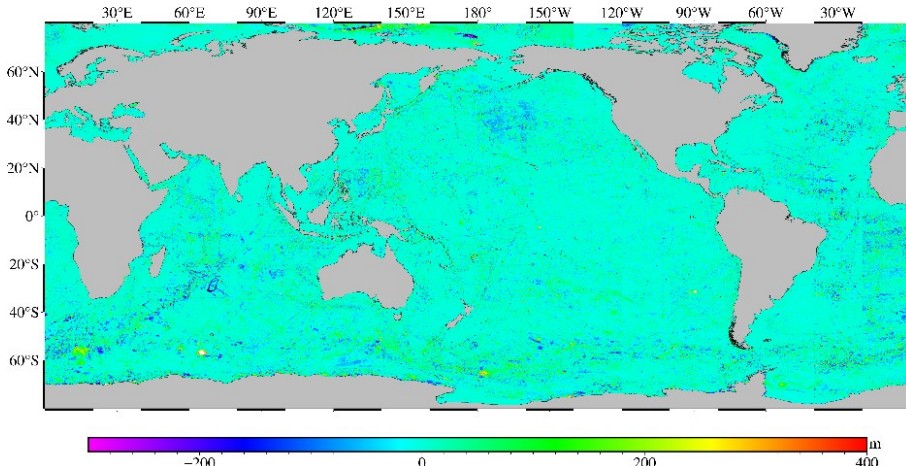

**Figure 3. Difference of seafloor topography between SDUST2023BCO model and topo_25.1 model.**

Using Eq. (6), the predicted bathymetry for each sub-region is obtained. In the overlapping areas of sub-regions, the final bathymetric value is obtained by averaging the values from these regions. Finally, the STUST2023BCO model is constructed using this method, as shown in Fig. 4.



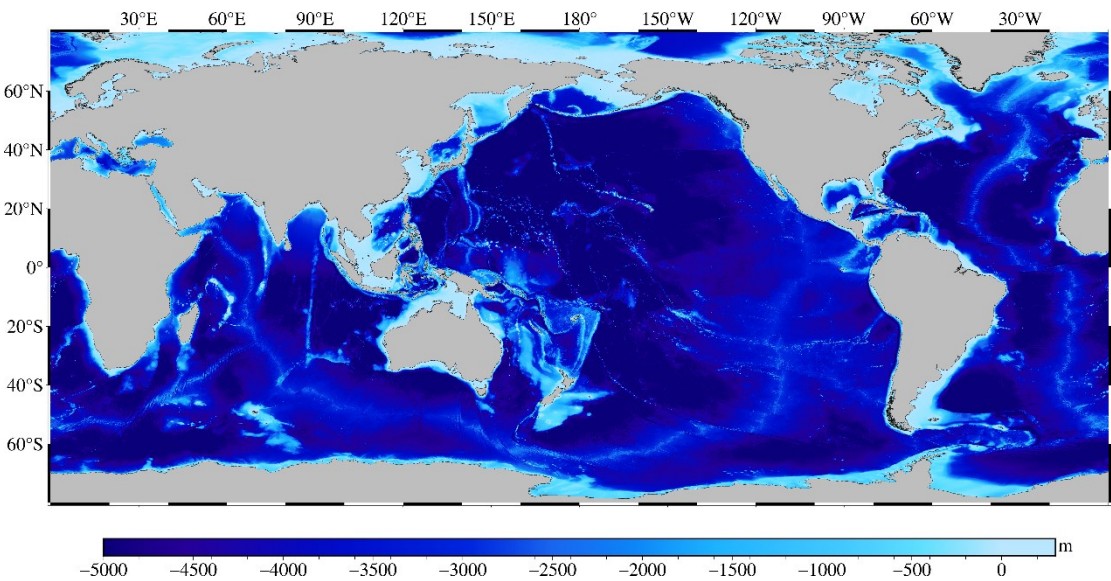

**Figure 4. The SDUST2023BCO model.**

## 4.3 Comparison with NCEI shipborne single-beam bathymetric points

The distribution of shipborne single-beam bathymetric points is showed in Fig. 1. The precision of the SDUST2023BCO model can be analyzed by using the shipborne single-beam bathymetric data from NCEI. In order to compare the precision of the SDUST2023BCO, GEBCO_2023, and topo_25.1 models, the RMS of the differences between the NCEI shipborne

single-beam bathymetric data and the three global marine bathymetric models is calculated within each sub-region, as shown in Table 3.

**Table 3. RMSE of differences between bathymetric model and shipborne single-beam bathymetric data (unit: m).**

| Aera | Evaluation Metrics | LAT1 | LAT2 | LAT3 | LAT4 | LAT5 | LAT6 | LAT7 | LAT8 |
|---|---|---|---|---|---|---|---|---|---|
| **LON1** | SDUST2023BCO | 65.55 | 54.85 | 110.00 | 53.00 | 41.60 | 35.53 | 63.36 | 30.59 |
|  | GEBCO_2023 | 83.02 | 57.03 | 114.81 | 115.66 | 55.38 | 50.40 | 101.87 | 48.29 |
|  | topo_25.1 | 80.47 | 59.28 | 117.69 | 72.83 | 43.94 | 47.46 | 62.92 | 36.71 |
| **LON2** | SDUST2023BCO | 9.44 | 54.94 | 102.50 | 89.41 | 62.88 | 122.41 | 85.33 | 120.73 |
|  | GEBCO_2023 | 16.20 | 67.03 | 105.12 | 95.95 | 42.96 | 128.78 | 104.80 | 135.55 |
|  | topo_25.1 | 14.68 | 68.64 | 109.62 | 102.69 | 128.41 | 127.38 | 90.25 | 136.13 |
| **LON3** | SDUST2023BCO | 11.88 | 44.10 | 84.40 | 75.13 | 53.00 | 165.52 | 68.91 | 68.27 |
|  | GEBCO_2023 | 16.81 | 35.86 | 90.38 | 88.05 | 64.27 | 171.80 | 86.21 | 79.43 |
|  | topo_25.1 | 13.71 | 45.10 | 88.88 | 87.53 | 66.45 | 172.36 | 74.00 | 78.51 |
| **LON4** | SDUST2023BCO | 19.46 | - | 53.34 | 53.56 | 69.81 | 138.95 | 75.72 | 40.12 |
|  | GEBCO_2023 | 32.68 | - | 74.55 | 66.98 | 74.97 | 129.62 | 68.17 | 61.79 |
|  | topo_25.1 | 20.55 | - | 66.39 | 65.49 | 71.83 | 192.42 | 119.28 | 49.32 |
| **LON5** | SDUST2023BCO | 13.54 | - | 69.08 | 50.83 | 56.58 | 146.90 | 53.62 | 44.35 |



| | | | | | | | | | |
|---|---|---|---|---|---|---|---|---|---|
| | GEBCO_2023 | 23.87 | - | 77.11 | 62.75 | 50.91 | 83.22 | 53.91 | 64.55 |
| | topo_25.1 | 14.05 | - | 75.52 | 62.14 | 64.24 | 164.16 | 60.54 | 54.11 |
| **LON6** | SDUST2023BCO | 1.55 | - | 39.23 | 89.20 | 73.58 | 59.11 | 76.50 | 37.05 |
| | GEBCO_2023 | 2.63 | - | 36.86 | 96.05 | 78.76 | 96.36 | 77.72 | 48.90 |
| | topo_25.1 | 2.27 | - | 47.91 | 121.88 | 95.89 | 90.65 | 78.35 | 45.80 |
| **LON7** | SDUST2023BCO | 45.29 | 73.18 | 75.40 | 73.14 | 37.94 | 45.14 | 79.51 | 62.62 |
| | GEBCO_2023 | 45.48 | 91.73 | 74.35 | 89.39 | 97.51 | 84.20 | 81.37 | 88.91 |
| | topo_25.1 | 31.04 | 84.75 | 77.33 | 85.89 | 68.72 | 77.76 | 86.80 | 74.45 |
| **LON8** | SDUST2023BCO | 50.40 | 70.51 | 110.90 | 108.05 | 67.89 | 46.67 | 63.41 | 51.28 |
| | GEBCO_2023 | 129.65 | 112.30 | 100.51 | 90.08 | 70.66 | 68.40 | 95.46 | 59.08 |
| | topo_25.1 | 83.81 | 94.38 | 115.30 | 118.94 | 77.84 | 64.56 | 77.85 | 60.65 |
| **LON9** | SDUST2023BCO | 42.35 | 63.25 | 60.33 | 76.69 | 94.14 | 52.94 | 53.10 | 57.51 |
| | GEBCO_2023 | 74.78 | 92.40 | 69.72 | 94.11 | 97.61 | 60.76 | 66.73 | 68.50 |
| | topo_25.1 | 75.94 | 77.61 | 73.69 | 98.14 | 103.83 | 62.57 | 73.73 | 65.00 |
| **LON10** | SDUST2023BCO | 86.58 | 75.64 | 107.87 | 150.55 | 170.00 | 78.29 | 51.85 | 68.37 |
| | GEBCO_2023 | 94.68 | 96.65 | 109.06 | 85.64 | 93.06 | 76.22 | 69.46 | 83.28 |
| | topo_25.1 | 94.04 | 90.29 | 112.11 | 231.69 | 209.53 | 88.53 | 67.60 | 78.56 |
| **LON11** | SDUST2023BCO | 98.10 | 67.24 | 86.15 | 84.09 | 103.52 | 60.85 | 79.76 | 67.20 |
| | GEBCO_2023 | 112.28 | 76.59 | 85.92 | 84.45 | 75.32 | 61.94 | 90.36 | 105.21 |
| | topo_25.1 | 113.42 | 76.37 | 86.99 | 87.22 | 114.59 | 70.45 | 88.06 | 101.02 |
| **LON12** | SDUST2023BCO | 68.21 | 68.76 | 62.98 | 63.68 | 51.63 | 56.84 | 82.77 | 45.54 |
| | GEBCO_2023 | 72.42 | 70.47 | 60.91 | 58.06 | 48.76 | 59.37 | 98.29 | 71.07 |
| | topo_25.1 | 66.67 | 70.87 | 63.21 | 70.02 | 58.34 | 68.04 | 90.03 | 55.38 |
| **LON13** | SDUST2023BCO | 24.92 | - | 70.83 | 92.18 | 54.72 | 114.17 | 65.85 | 33.06 |
| | GEBCO_2023 | 45.05 | - | 75.44 | 90.76 | 44.56 | 59.65 | 80.47 | 39.98 |
| | topo_25.1 | 39.73 | - | 71.94 | 97.40 | 60.02 | 124.54 | 74.88 | 41.57 |
| **LON14** | SDUST2023BCO | 48.71 | 132.98 | 48.73 | 77.96 | 58.43 | 80.05 | 73.78 | 31.75 |
| | GEBCO_2023 | 50.71 | 133.43 | 48.56 | 78.16 | 59.94 | 77.68 | 84.25 | 58.81 |
| | topo_25.1 | 49.73 | 132.84 | 51.05 | 81.37 | 66.49 | 85.88 | 82.20 | 40.26 |
| **LON15** | SDUST2023BCO | 156.79 | 59.05 | 96.16 | 90.32 | 56.70 | 111.03 | 92.13 | 62.90 |
| | GEBCO_2023 | 164.46 | 57.94 | 95.50 | 98.09 | 67.53 | 119.65 | 101.24 | 62.34 |
| | topo_25.1 | 162.80 | 61.01 | 98.99 | 98.82 | 66.88 | 119.96 | 100.57 | 78.95 |
| **LON16** | SDUST2023BCO | 48.93 | 45.00 | 112.02 | 86.95 | 65.75 | 40.90 | 70.13 | 60.13 |
| | GEBCO_2023 | 67.81 | 50.83 | 114.80 | 96.41 | 80.22 | 23.03 | 91.02 | 52.04 |
| | topo_25.1 | 64.64 | 50.82 | 116.23 | 94.84 | 79.69 | 52.08 | 77.58 | 69.75 |
| **LON17** | SDUST2023BCO | 43.29 | 94.30 | 129.94 | 114.02 | 103.47 | 40.62 | 110.35 | 42.10 |
| | GEBCO_2023 | 44.02 | 106.14 | 135.38 | 129.15 | 108.97 | 46.61 | 167.49 | 66.70 |
| | topo_25.1 | 58.87 | 98.55 | 134.60 | 126.66 | 163.78 | 58.29 | 154.82 | 50.35 |
| **LON18** | SDUST2023BCO | 60.85 | 71.59 | 68.84 | 64.33 | 95.16 | 82.34 | 173.11 | 38.66 |
| | GEBCO_2023 | 80.66 | 79.93 | 78.35 | 79.73 | 97.81 | 116.30 | 181.99 | 50.91 |
| | topo_25.1 | 73.72 | 73.54 | 78.17 | 72.96 | 91.39 | 92.75 | 172.09 | 47.96 |



Table 3 shows that the SDUST2023BCO, topo_25.1, and GEBCO_2023 models have their strengths and weaknesses in different sub-regions. The SDUST2023BCO model demonstrates superior precision compared to the GEBCO_2023 and
topo_25.1 models, achieving higher accuracy in 112 and 134 sub-regions, respectively, which corresponds to approximately 80.00% and 95.71% of the total sub-regions. Considering all indicators, the SDUST2023BCO model shows higher accuracy than the GEBCO_2023 and topo_25.1 models.

To validate the accuracy of the SDUST2023BCO, GEBCO_2023, and topo_25.1 models, each model is interpolated onto all shipborne single-beam bathymetric points using a cubic spline interpolation method, the relevant statistical results are
showed in Table 4. Table 4 shows that the models ranked from highest to lowest accuracy are the SDUST2023BCO model, the GEBCO_2023 model, and the topo_25.1 model. Compared to the GEBCO_2023 and topo_25.1 models, the accuracy of the SDUST2023BCO model is improved by 0.28m and 15.57m, respectively. The statistical results show that the SDUST2023BCO model exhibits superior accuracy compared to the GEBCO_2023 and topo_25.1 models, aligning more closely with the shipborne single-beam bathymetric data.

**Table 4. Statistical results between SDUST2023BCO, GEBCO_2023 and topo_25.1 models and shipborne single-beam bathymetric points (unit: m).**

| Model | Max | Min | Mean | STD | RMS |
|---|---|---|---|---|---|
| SDUST2023BCO | 1846.19 | -1782.62 | 8.53 | 90.23 | 90.63 |
| GEBCO_2023 | 4413.78 | -2981.52 | 10.53 | 90.51 | 91.12 |
| topo_25.1 | 977.65 | -977.65 | 9.35 | 105.80 | 106.21 |

Figure 5 shows the histogram distribution of the differences between the SDUST2023BCO, GEBCO_2023, and topo_25.1 models and the shipborne single-beam bathymetric data, showing that the error distributions of all three models exhibit a normal distribution. The percentages of differences between the bathymetric models and the actual bathymetry falling within
the ±50m range are 72.44%, 72.01%, and 68.92%, respectively. The distribution of differences between the SDUST2023BCO model and the shipborne single-beam bathymetric data is more concentrated, demonstrating a superior ability to reflect the information of the seafloor topography.

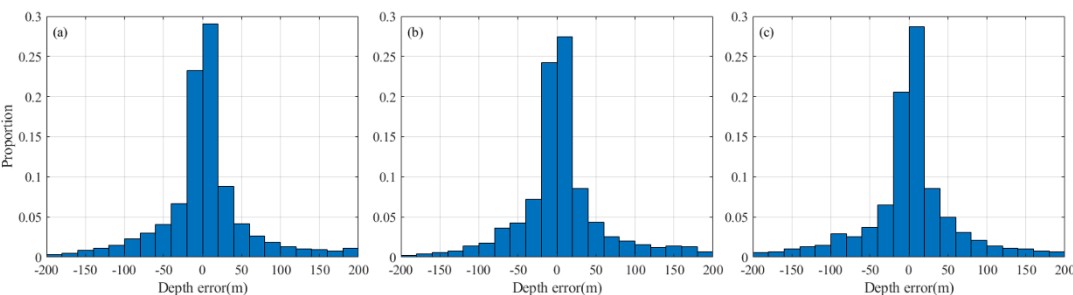

**Figure 5. Distribution histogram of the difference between SDUST2023BCO, topo_25.1, GEBCO_2023 models and shipborne**
**single-beam bathymetric data.**




Table 5 presents the statistical results of the comparison between the SDUST2023BCO, GEBCO_2023, and topo_25.1 models and shipborne single-beam bathymetric data at different depths. The accuracy of the SDUST2023BCO model outperforms the GEBCO_2023 and topo_25.1 models across depth intervals of -1000m to -2000m, -2000m to -3000m, and below -3000m, with improvements of 8.05m and 17.04m, 7.71m and 9.25m, and 0.97m and 10.41m, respectively. In waters shallower than 1000m, the SDUST2023BCO model demonstrates a precision enhancement of 29.44m compared to the topo_25.1 model. Overall, the SDUST2023BCO model exhibits commendable accuracy in deeper waters, with potential for enhanced precision in shallower areas.

**Table 5. Statistical results of the difference between SDUST2023BCO, GEBCO_2023 and topo_25.1 models and the measured bathymetry at shipborne single-beam bathymetric points in different ranges of bathymetry (Unit: m).**

| Different ranges of bathymetry | Number of points | SDUST2023BCO | | GEBCO_2023 | | topo_25.1 | |
|---|---|---|---|---|---|---|---|
| | | STD | RMS | STD | RMS | STD | RMS |
| (0, -1000) | 26753617 | 87.18 | 87.63 | 78.16 | 79.31 | 116.62 | 116.83 |
| [-1000, -2000) | 10794924 | 84.58 | 84.71 | 92.63 | 92.98 | 101.62 | 101.80 |
| [-2000, -3000) | 15120787 | 77.88 | 78.63 | 85.59 | 86.35 | 87.13 | 88.09 |
| [-3000, -∞) | 59649674 | 95.31 | 95.69 | 96.28 | 96.75 | 105.72 | 106.19 |

## 4.4 Comparison with SIO topo_25.1 and GEBCO_2023

To verify the accuracy of the SDUST2023BCO model, the bathymetry of the SDUST2023BCO, GEBCO_2023, and topo_25.1 models are respectively calculated, as shown in Table 6.

**Table 6. Statistical results of bathymetry of SDUST2023BCO, GEBCO_2023 and topo_25.1 models (unit: m).**

| Model | Max | Min | Mean | STD | RMS |
|---|---|---|---|---|---|
| SDUST2023BCO | 0 | -10869.8 | -3476.2 | 1749.4 | 3892.6 |
| GEBCO_2023 | 0 | -10874.1 | -3479.1 | 1750.9 | 3894.9 |
| topo_25.1 | 0 | -10804.8 | -3478.0 | 1749.4 | 3893.2 |

Table 6 shows that the STD of the SDUST2023BCO model is 1749.4m, differing by 1.5m and 0.0m from the GEBCO_2023 and topo_25.1 models, respectively. Additionally, the min and mean values of the SDUST2023BCO model are closely aligned with those of the GEBCO_2023 and topo_25.1 models. Considering all these indicators, the consistency of the SDUST2023BCO model with the GEBCO_2023 and topo_25.1 models is effectively validated.

To further validate the consistency of the SDUST2023BCO model with other models, the differences between the SDUST2023BCO, GEBCO_2023, and topo_25.1 models are calculated. Relevant statistical outcomes are showed in Table 7. Owing to the SDUST2023BCO model having a resolution of 1′×1′, the bathymetric values at 1′ grid nodes are selected from the GEBCO_2023 model, and the GEBCO_2023 model is processed into a bathymetric model with a resolution of 1′.

**Table 7. Statistical results of differences of SDUST2023BCO, GEBCO_2023 and topo_25.1 models (unit: m)**



| Model | Max | Min | Mean | STD | RMS |
|---|---|---|---|---|---|
| SDUST2023BCO - GEBCO_2023 | 4316.1 | -4043.2 | 3.0 | 58.4 | 58.5 |
| SDUST2023BCO- topo_25.1 | 2308.8 | -3788.4 | 1.8 | 41.2 | 41.2 |
| GEBCO_2023 - topo_25.1 | 5204.6 | -5219.7 | 1.1 | 70.3 | 70.3 |

Table 7 shows that the STD of the differences between the SDUST2023BCO model and the other models are 58.4m and 41.2m, respectively. This indicates that the SDUST2023BCO model has the highest correlation with the topo_25.1 model, followed by the GEBCO_2023 model. The SDUST2023BCO model shows commendable consistency with the GEBCO_2023 and topo_25.1 models, demonstrating the reliability and effectiveness of this method.

Figure 6 shows the histogram distributions of the differences between the three bathymetric models. From Fig. 6(a), the differences between the SDUST2023BCO and GEBCO_2023 models are mainly within the range of -100m to 100m, accounting for approximately 94.51%. From Fig. 6(b), the differences between the SDUST2023BCO and topo_25.1 models within the same range account for about 96.89%. From Fig. 6(c), the differences between the topo_25.1 model and the GEBCO_2023 model within the range of -100m to 100m account for approximately 93.38%. Based on the above statistics, the SDUST2023BCO model exhibits commendable consistency with the GEBCO_2023 and topo_25.1 models.

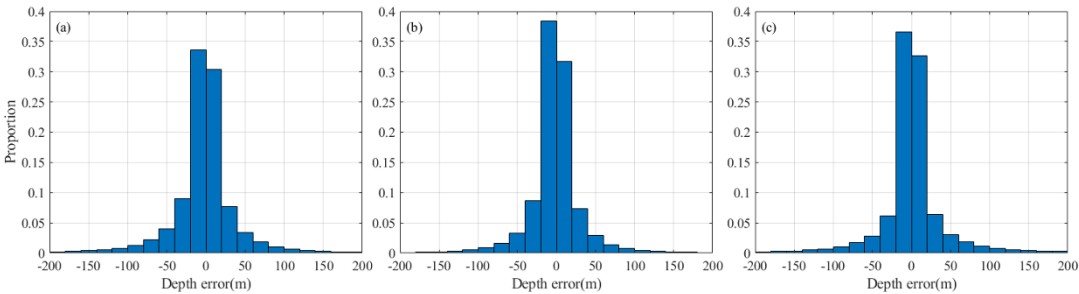

**Figure 6. histogram of the difference between SDUST2023BCO, topo_25.1 and GEBCO_2023 models.**

According to the law of error propagation, assuming that the SDUST2023BCO, GEBCO_2023, and topo_25.1 models are uncorrelated, the STD of these three models can be expressed as:

$$\begin{cases} STD^2_{S\_G} = STD^2_S + STD^2_G \\ STD^2_{S\_t} = STD^2_S + STD^2_t \\ STD^2_{G\_t} = STD^2_G + STD^2_t \end{cases} \tag{7}$$



where $STD_{S\_G}$, $STD_{S\_t}$ and $STD_{G\_t}$ respectively represent the STD of comparisons between the SDUST2023BCO and GEBCO_2023 models, the SDUST2023BCO and topo_25.1 models, and the GEBCO_2023 and topo_25.1 models. $STD_S$, $STD_G$ and $STD_t$ respectively represent the STD of the bathymetric values of the SDUST2023BCO, GEBCO_2023, and topo_25.1 models.

Using Eq. (7) and the statistical results in Table 7, the STD of the bathymetry for the SDUST2023BCO, GEBCO_2023, and topo_25.1 models can be calculated as 9.11m, 57.69m, and 40.18m, respectively. This result indicates that the precision of the three models, from highest to lowest, is the SDUST2023BCO, topo_25.1, and GEBCO_2023 models. This effectively demonstrates that the SDUST2023BCO model has the highest accuracy among the three models.

Furthermore, four regions are selected to validate the accuracy of bathymetric model, specifically the North Pacific Ocean (120°E-120°W, 0°N-65°N), the South Pacific Ocean (120°E-80°W, 80°S-0°S), the Atlantic Ocean (0°W-60°W, 80°S-80°N), and the Indian Ocean (0°E-60°E, 80°S-30°N). Relevant statistical results are showed in Table 8. Table 8 shows that the SDUST2023BCO model exhibits better accuracy across all regions, further substantiating its reliability in the various oceans.

**Table 8. Statistical results of the differences between the SDUST2023BCO and GEBCO_2023 and topo_25.1 models within various regions (unit: m).**

| Range | Model | Max | Min | Mean | STD | RMS |
|---|---|---|---|---|---|---|
| North Pacific Ocean (120°E-120°W, 0°N-65°N) | SDUST2023BCO- GEBCO_2023 | 2604.23 | -2791.71 | -1.96 | 55.43 | 55.46 |
| | SDUST2023BCO- topo_25.1 | 1290.71 | -1489.07 | -1.47 | 36.09 | 36.12 |
| | GEBCO_2023- topo_25.1 | 3992.29 | -3301.33 | -0.48 | 51.55 | 51.55 |
| South Pacific Ocean (120°E-80°W, 80°S-0°S) | SDUST2023BCO- GEBCO_2023 | 4043.23 | -4316.07 | -4.69 | 62.08 | 62.26 |
| | SDUST2023BCO- topo_25.1 | 2551.56 | -1805.10 | -2.71 | 39.12 | 39.21 |
| | GEBCO_2023- topo_25.1 | 5219.67 | -4996.58 | -1.93 | 73.79 | 73.81 |
| Atlantic Ocean (0°W-60°W, 80°S-80°N) | SDUST2023BCO- GEBCO_2023 | 2413.54 | -3012.18 | -3.95 | 60.60 | 60.72 |
| | SDUST2023BCO- topo_25.1 | 3788.43 | -1685.94 | -0.40 | 43.04 | 43.04 |
| | GEBCO_2023- topo_25.1 | 5204.59 | -2595.10 | 3.55 | 73.76 | 73.85 |
| Indian Ocean (0°E-60°E, 80°S-30°N) | SDUST2023BCO- GEBCO_2023 | 2477.18 | -2305.65 | -3.57 | 58.75 | 58.83 |
| | SDUST2023BCO- topo_25.1 | 1686.75 | -2308.84 | -0.52 | 46.36 | 46.36 |
| | GEBCO_2023- topo_25.1 | 2633.55 | -3212.54 | 3.04 | 75.25 | 75.31 |

**4.5 Data availability**

The global bathymetric model (SDUST2023BCO) can be downloaded at https://doi.org/10.5281/zenodo.13341896 (Zhou et al., 2024). The dataset includes geospatial information (latitude, longitude) and corresponding bathymetric values.

**5. Conclusion**

Considering the effectiveness in the construction of bathymetric models, the influence of long-wavelength information derived from multi-source geodetic datasets, and the nonlinear interrelation between multi-source marine geodetic data and bathymetry, a new global marine model, designated as SDUST2023BCO model, has been constructed. This model has a

resolution of 1×1', with spatial coverage ranging from 0° to 360°E in longitude and from 80°S to 80°N in latitude. This model is constructed based on the MLP neural network, integrating the differences from multi-source marine geodetic data.

The accuracy of the SDUST2023BCO model has been evaluated by using shipborne single-beam bathymetric data, as well as the GEBCO_2023 and topo_25.1 models.

Compared to the shipborne single-beam bathymetric data, the SDUST2023BCO model achieves an accuracy of 90.23m, which is superior to other bathymetric models, demonstrating the reliability of the SDUST2023BCO model. Through the comparison of the accuracy of three models in different depth, the SDUST2023BCO model demonstrates superior accuracy

in deeper water regions.

The discrepancies between the SDUST2023BCO model and the GEBCO_2023, topo_25.1 models primarily fall within ±100m, confirming the consistency of the SDUST2023BCO model with existing models. This paper also evaluates the accuracy of the SDUST2023BCO model in four distinct regions across the Pacific, Atlantic, and Indian Oceans, effectively validating its reliability.

All these verifications show that SDUST2023BCO reaches an international advanced level of global bathymetric models. The accuracy of SDUST2023BCO model is better than that of GEBCO_2023 and topo_25.1 models, especially in deeper water regions.

**Author contribution.** SZ presented the algorithm and carried out the experimental results. JG put forward the idea and polished the entire manuscript. HZ and YJ downloaded all products in this work and polished the manuscript. HS contributed

to the validation with NCEI shipborne single-beam bathymetric points. XL contributed to the validation with SIO topo_25.1 and GEBCO_2023. All authors checked and gave related comments for this work.

**Competing interests.** All authors have no competing interests to declare that are relevant to the content of this article.

**Acknowledgments.** We express our gratitude to the SIO and GEBCO for providing bathymetric models, the NCEI for supplying shipborne single-beam bathymetric data, the SIO for providing vertical deflection and vertical gravity gradient

data, the Shandong University of Science and Technology for providing gravity anomaly data, and the Centre National d'Etudes Spatiales for offering the MDT model. Lastly, we would also like to thank the creators and contributors of the plotting tools.

**Financial support.** This study is supported by the National Natural Science Foundation of China (grant Nos. 42274006 and 42192535).

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
