# Peer review of "SDUST2023BCO: a global seafloor model determined from multilayer perceptron neural network using multi-source differential marine geodetic data"

_Earth System Science Data, 2024_

## Author Response (AR1)

**Dear editor and reviewer:**

Thanks for your suggestions and comments. We have carefully revised the manuscript according to the comments. Your opinions are reasonable, greatly helping me improve my article. We respond to the review comments point by point. The response to the reviewer's comments is as follows:

**Reviewer #1:**

**Point 1. Validation Approach**: Since the shipborne single-beam bathymetric data and the topo_25.1 model were used during the training process, it is expected that the derived bathymetric model will align well with them. This raises concerns about the validity of the comparison results, as they cannot serve as an independent assessment of the model's accuracy. Only independent datasets that were not used in the training process can provide a proper validation of the bathymetric model.

**Response:** Thanks for your suggestions and comments. Your opinions greatly helping me improve our article.

In this paper, all available shipborne single-beam bathymetric data are used as the training set for the MLP neural network. There are some problems such as uneven distribution and small quantity of shipborne single-beam bathymetric data, especially in high latitude areas. In this paper, all available shipborne single-beam bathymetric data are used to improve the accuracy of SDUST2023BCO model.

The use of shipborne single-beam bathymetric data served as a validation measure for the SDUST2023BCO model, aiming to demonstrate its alignment with shipborne single-beam bathymetric data. Based on expert's suggestion, the expression "The SDUST2023BCO model achieves the highest accuracy" is changed to "The SDUST2023 model exhibits closer resemblance to shipborne single-beam bathymetric data", thus perfecting this expression.

According to the research method of Zhou et al. (2024), which is also the work of our team, a global high-precision bathymetric model is constructed. In this paper, the Caribbean Sea is taken as the experimental area, and the bathymetric model of this area is constructed by the same method. Compared with the single ship-borne bathymetric check points, the feasibility and effectiveness of this method are effectively verified.

To validate the accuracy of the SDUST2023BCO model, Sect. 4.4 of this paper presents a comparative analysis between the SDUST2023BCO model, GEBCO_2023 and topo_25.1. This analysis effectively demonstrates the consistent accuracy of the SDUST2023BCO model.

In conclusion, the phrasing of the article is revised, with the changes clearly marked in red within the manuscript.

**Point 2. Machine Learning Methodology**: A standard machine learning workflow typically involves dividing the data into training, validation, and test datasets. The training dataset is used to train the model, the validation dataset is used to fine-tune the hyperparameters and assess performance during training, and the test dataset serves as a completely separate subset to evaluate

the final model's performance. In this manuscript, I only see references to input and output data. Please clarify how your approach adheres to standard machine learning practices, and make adjustments if necessary.

**Response:** Thanks for your suggestions and comments. Your opinions greatly helping me improve my article.

First, we divided part of the dataset into training and validation sets to search for the optimal hyperparameters. Through training and validation, the hyperparameters were determined, which consist of four hidden layers with 512, 256, 128, and 64 neurons per layer, respectively, a learning rate of 0.0001, a batch_size of 8, and other details as outlined in Sect. 3.2. The hyperparameters determined by the training and validation sets can effectively ensure the training accuracy of the MLP neural network, as shown in Table 2 of the manuscript.

Secondly, after identifying the corresponding hyperparameters, the MLP neural network was trained using the same set of hyperparameters. To enhance the accuracy of training and prediction in specific sub-regions, the parameters in some sub-regions were fine-tuned. The training points consist of all shipborne single-beam bathymetric points, while the test data (prediction data) corresponds to the center of each $1'\times1'$ grid, as shown in Fig. 2 of the manuscript.

Finally, the predicted values at the prediction points were used to construct the seafloor topography model for each sub-region.

We revised Sect. 3.3 of the manuscript based on the suggestions, with the modifications as follows:

"The second step is to select appropriate neural network parameters. The choice of parameters is critical for the training and prediction accuracy of the MLP model. This includes the initialization of weights and biases, the number of hidden layers, activation function, learning rate, and batch size. In order to achieve high-precision in training and prediction, the selection of parameters may be adjusted in different sub-regions. Relevant parameters are initially set randomly, and then individual parameters are adjusted based on training accuracy until the most suitable parameters are obtained. For example, if the training accuracy is poor, increasing the number of hidden layers, the number of neurons in each hidden layer, or the number of iterations, or decreasing the learning rate or batch size can help achieve the most appropriate parameters. The relevant hyperparameters were determined through the training set and validation set. In this paper, a four-layer hidden neural network is used, with each layer containing 512, 256, 128, and 64 neurons, respectively. The learning rate is set to 0.0001, and the batch_size is set to 8."

**Point 3. Dataset Selection and Hyperparameter Tuning**: The rationale for selecting the specific datasets (gravity anomalies, vertical gravity gradient, vertical deflection, and MDT) for bathymetry modeling requires further explanation. Additionally, the criteria for choosing the hyperparameters in your machine learning model need to be detailed and justified.

**Response:** Thanks for your suggestions and comments. Your opinions greatly helping me improve my article.

**Dataset Selection**

Traditional methods for bathymetric inversion generally use gravity anomalies and vertical gravity gradients as initial data, so this paper uses these two types of data as input

data for the MLP neural network.

When calculating gravity anomalies and vertical gravity gradients, vertical deflection often is also considered. As a type of marine gravity data, vertical deflection can also provide bathymetric information (Annan and Wan, 2022). Therefore, this paper introduces vertical deflection data as part of the input data.

In Sect. 3.2, we described the selection of the dataset and highlighted the changes in red within the manuscript. The modifications are as follows:

"The organization format of input/output data significantly influences the training and predictive accuracy of MLP neural networks. In the traditional methods of constructing seafloor topography models, marine gravity data is typically used as the initial data. Based on the correlation between gravity data and bathymetry (Smith & Sandwell, 1994), the gravity anomaly, the meridional and prime components of vertical deflection are used as input data for training and prediction. Since MDT data can reflect the bathymetric information to a certain extent (Pujol et al., 2018; Mulet et al., 2021), MDT data has also been introduced."

**Hyperparameter Tuning**

The criteria for selecting hyperparameters were based on the training accuracy of the model. Currently, there is no clear method for hyperparameter selection; suitable hyperparameters can only be identified through continuous trial. However, some patterns emerge during this process. For example, when the training accuracy is low, increasing the number of hidden layers or neurons in each hidden layer, or decreasing the batch_size or learning rate. High-precision prediction results can only be achieved when the training accuracy of the MLP neural network reaches a certain level.

We have revised and added the explanation of the hyperparameter selection criteria in Sect. 3.3, with the changes marked in red in the manuscript. The specific revisions are as follows:

"The second step is to select appropriate neural network parameters. The choice of parameters is critical for the training and prediction accuracy of the MLP model. This includes the initialization of weights and biases, the number of hidden layers, activation function, learning rate, and batch size. In order to achieve high-precision in training and prediction, the selection of parameters may be adjusted in different sub-regions. Relevant parameters are initially set randomly, and then individual parameters are adjusted based on training accuracy until the most suitable parameters are obtained. For example, if the training accuracy is poor, increasing the number of hidden layers, the number of neurons in each hidden layer, or the number of iterations, or decreasing the learning rate or batch size can help achieve the most appropriate parameters. The relevant hyperparameters were determined through the training set and validation set. In this paper, a four-layer hidden neural network is used, with each layer containing 512, 256, 128, and 64 neurons, respectively. The learning rate is set to 0.0001, and the batch_size is set to 8."

**Point 4. Role of MDT Data**: The authors claim that MDT data reflects variations in bathymetric models, justifying its inclusion in the model alongside other datasets. The logic here is unclear. Does this imply that variations in seafloor topography need to be considered? What specific contribution does MDT data make to the recovery of seafloor topography?

**Response:** Thanks for your suggestions and comments. Your opinions greatly helping me

improve my article.

We apologize for the unclear statement. The reason for introducing the MDT model in this paper is that the MDT model can provides bathymetric information to a certain extent. We have revised this section and added relevant references. The modifications are as follows:

"Since MDT data can reflect the bathymetric information to a certain extent (Pujol et al., 2018; Mulet et al., 2021), MDT data has also been introduced."

The impact of MDT data on the accuracy of seafloor topography models was studied in the Caribbean Sea (17-22°N, 271-280°W). Two seafloor topography models were constructed using this method described in this paper: Model1 and Model2. When constructing the Model1 model, MDT data was used; however, MDT data was not used in the construction of the Model2 model. The methods used to construct Model1 and Model2 are consistent with the method developed in this paper. Both Model1 and Model2 were interpolated to the shipborne single-beam bathymetric points using cubic spline interpolation, and the relevant statistical results are shown in Table 1.

**Table 1: Statistical results of the differences between Model1 and Model2 and shipborne single-beam bathymetric points (unit: m)**

| Model | Max | Min | Mean | STD | RMS |
|-------|-----|-----|------|-----|-----|
| Model1 | 799.25 | -807.76 | 2.54 | 78.62 | 78.66 |
| Model2 | 809.26 | -806.11 | 3.56 | 82.51 | 82.59 |

As shown in Table 1, the accuracy of Model1, constructed using MDT data, improved by approximately 3.89m compared to the accuracy of Model2. This effectively validates that MDT data can enhance the accuracy of seafloor topography models.

In conclusion, the MDT data plays a vital role in the accuracy of the seafloor topography model. Using MDT data can effectively improve the accuracy of the SDUST2023BCO model. We have made revisions in the text and highlighted them in red within the manuscript.

**Point 5. Equation (7)**: L315, Equation (7) presents the standard deviation (STD) of the differences between two models on the left side and the STD of the bathymetric values of each model on the right side. However, none of these terms represent errors, and therefore, error propagation cannot be applied. The error estimation in this context is invalid.

**Response:** Thanks for your suggestions and comments. Your opinions greatly helping me improve my article.

We believe that this method can be applied. This paper uses seafloor topography models from two different institutions (GEBCO_2023, topo_25.1) and assumes that there is no correlation between the three models. When applying the law of error propagation, $STD_{S\_G}$, $STD_{S\_t}$ and $STD_{G\_t}$ represent the STD of the differences between these two models. The difference between the two models can be considered as the error between them, and through this theory, the corresponding the STD of bathymetric model can be determined. We believe that this method offers an alternative perspective for analyzing the accuracy of the SDUST2023BCO model, thereby validating its reliability.

This paper references the method used in Yuan et al. (2023), which employed the same

approach to verify the accuracy of the constructed model.

In conclusion, we believe that the law of error propagation can be used to verify the accuracy of the SDUST2023BCO model.

**Point 6. Accuracy vs. Precision**: Throughout the manuscript, the terms "accuracy" and "precision" are used interchangeably. It seems the authors may not fully understand the distinction between these two concepts. Please revise the manuscript to ensure the correct use of these terms.

**Response:** Thanks for your suggestions and comments. Your opinions greatly helping me improve my article.

We carefully reviewed the use of "accuracy" and "precision" in the manuscript, analyzing their application through literature review. We standardized the terminology to "accuracy" throughout the text, with these changes highlighted in red within the manuscript. Additionally, we revised certain expressions to meet publication requirements, and these revisions are also marked in red within the manuscript.

**Point 7**. Lastly, the overall clarity of the writing needs improvement. To justify publication, a thorough revision and reformulation of several sections are required.

**Response:** Thanks for your suggestions and comments. Your opinions greatly helping me improve my article.

We are so sorry for our incorrect writing. We tried our best to improve the manuscript and made some changes in the manuscript. These changes will not influence the content and framework of the paper. And we did not list the changes but marked in red within the manuscript.

**Other comments/questions:**

1. L15, The multisource differential marine geodetic data. The term "differential" is unclear in this context. Do you mean "different"? Please clarify the expression.

**Response:** Thanks for your suggestions and comments. Your opinions greatly helping me improve my article.

We apologize for the ambiguity in our previous expression. The term "differential" refers to the difference between the data at the grid points and the data at the training or prediction points. We have revised this expression, and the modifications are as follows:

"The multi-source marine geodetic data used include gravity anomaly data released by Shandong University of Science and Technology, vertical gravity gradient and the vertical deflection data released by Scripps Institution of Oceanography, as well as mean dynamic topography data released by the Centre National d'Etudes Spatiales."

2. L48, "the Gulf of Guinea (Annan and Wan, 2020)," This reference is missing from the bibliography. Please verify the inclusion and accuracy of all cited references.

**Response:** Thanks for your suggestions and comments. Your opinions greatly helping me improve my article.

We have carefully verified the accuracy of the references and corrected citation issues in certain parts of the manuscript. At the same time, we also reviewed the alignment between the references and their application, with the modified content highlighted in red within the

manuscript.

3. L53, Consequently, a global bathymetric model can be constructed by integrating the nonlinear components inherent in the relationship between multi-source marine geodetic data and the seafloor topography while accounting for long-wavelength information present in these datasets." Why is long-wavelength information being emphasized here? How it can be enhanced? The authors do not provide a clear rationale or analysis here.

**Response:** Thanks for your suggestions and comments. Your opinions greatly helping me improve my article.

We apologize for the ambiguity in our previous expression. The purpose of this paper is to use the MLP neural network to integrate the multi-source differential marine geodetic data to construct a global bathymetric model. Using the multi-source differential marine geodetic data, rather than multi-source marine geodetic data, can help attenuate the long-wavelength information present in the multi-source marine geodetic data. This long-wavelength information plays a negative role in the inversion of the bathymetric model. At the same time, we have revised this sentence as follows:

"Consequently, a global bathymetric model can be constructed by integrating the nonlinear components inherent in the relationship between multi-source marine geodetic data and the seafloor topography. At the same time, the long-wavelength information in multi-source marine geodetic data affects the prediction accuracy of the seafloor topography model. Therefore, it is necessary to mitigate the impact of long-wavelength information on the model's accuracy."

4. L60, However, this method may introduce bias when processing input data with convolutional layers. How exactly does this method introduce bias? More specific details are needed.

**Response:** Thanks for your suggestions and comments. Your opinions greatly helping me improve my article.

When convolving with a convolution kernel, it can lead to distortion of some data, thereby affecting the accuracy of the bathymetric model constructed. We have revised the sentence based on the expert's advice, and the modifications are highlighted in red within the manuscript.

5. L70, network, integrating the differences from multi-source marine geodetic data (gravity anomalies, vertical gravity gradients, the meridional and prime components of vertical deflection, mean dynamic topography). The phrase "integrating the differences" is unclear. What differences are being integrated? Please explain.

**Response:** Thanks for your suggestions and comments. Your opinions greatly helping me improve my article.

We revised this sentence, and the revised sentence is as follows:

"This model is constructed based on a MLP neural network, using the differences between the multi-source marine geodetic data (gravity anomalies, vertical gravity gradients, the meridional and prime components of vertical deflection, mean dynamic topography) of training/prediction points, and their surrounding grid points."

6. L115, The residual shipborne single-beam bathymetric points. What does "residual" refer to here? How are these residuals computed, and why are they used?

**Response:** Thanks for your suggestions and comments. Your opinions greatly helping me improve my article.

We apologize for the ambiguity in our previous expression. "The residual shipborne single-beam bathymetric points" are all the shipboard single-beam bathymetric points remaining after the process described in Sect. 2.1. We revised this sentence, and the revised sentence is as follows:

"The 112319002 shipborne single-beam bathymetric points are used to train the MLP model which is employed to construct the SDUST2023BCO model."

7. L120. The precision of the SDUST2022GRA model in local coastal and high-latitude areas has been effectively enhanced. It is stated that the precision has been effectively enhanced. Enhanced compared to what? A baseline or previous model?

**Response:** Thanks for your suggestions and comments. Your opinions greatly helping me improve my article.

We revised this sentence, and the revised sentence is as follows:

"In local coastal and high-latitude regions, SDUST2022GRA showed an enhancement of 0.16-0.24 mGal compared to the altimeter-derived global gravity anomaly models (DTU17, V32.1, NSOAS22) and shipborne gravity measurements."

8. L125, This version of the gravity models, which determines from Cryosat LRM, Altika, Cryosat-SAR, and Sentinel-3A/B, has high quality and precision. Please provide citations for these models and include the precision/accuracy metrics.

**Response:** Thanks for your suggestions and comments. Your opinions greatly helping me improve my article.

SIO has been dedicated to constructing a high-precision global marine gravity model, achieving significant results that have gained international recognition. Each year, SIO updates and optimizes the global marine gravity model based on newly acquired satellite altimetry data. The version 32.1 of gravity model used in this paper has shown promising results worldwide; however, no research has yet been conducted to validate its accuracy.

To avoid ambiguity, we made some modifications of this sentence while keeping its original meaning. The revised version is as follows:

"Based on the correlation between vertical deflection, vertical gravity gradient, and bathymetry, those data can also be utilized to predict bathymetry. These gravity data are derived from the 32.1 version released by SIO in 2022, with a resolution of 1'×1', and can be freely obtained from https://topex.ucsd.edu/pub/global_grav_1min/."

9. L130, foundational data. Not a clear expression. The term "foundational data" is vague. Please clarify what is meant by this.

**Response:** Thanks for your suggestions and comments. Your opinions greatly helping me improve my article.

We revised this sentence, and the revised sentence is as follows:

"The MDT model has a resolution of 7.5'×7.5', and is calculated using data from the

CNES-CLS15 mean sea level model (Pujol et al., 2018), the GOCO05S geoid model, hydrographic data, and drifting data."

10. L185, slope. What role does slope play in seafloor topography recovery? Please provide a detailed explanation.

**Response:** Thanks for your suggestions and comments. Your opinions greatly helping me improve my article.

Slope, as a crucial information about seafloor topography, can reveal the changing trend of bathymetry at a point. We believe that, in constructing a seafloor topography model, slope can effectively reflect the variations in bathymetry, and thus slope is employed as an input data to construct the SDUST2023BCO model in this paper.

11. Fig. 2, Output data: topo_25.1 bathymetry-Shipborne single-beam bathymetry. Eq. (4). Does this mean you used both topo_25.1 bathymetry and Shipborne single-beam bathymetry as labels to train the model? Clarification is needed.

**Response:** Thanks for your suggestions and comments. Your opinions greatly helping me improve my article.

We have revised Fig. 2 to better illustrate the format of the output data, with the revisions as follows:

[Figure]

**Figure 2. The organization of input/output data and structure of MLP.**

12. Table 2, $R^2$? Please define $R^2$ and provide the formula used to compute it.

**Response:** Thanks for your suggestions and comments. Your opinions greatly helping me improve my article.

We provided the calculation formula of $R^2$ before Table 2, which is modified as follows:

"The coefficient of determination ($R^2$) is introduced to evaluate the training accuracy of the MLP neural network, The calculation equation is as follows:

$$R^2 = (1 - \frac{(\sum_{i=1}^{n}(h_i^{pred} - h_i)^2}{\sum_{i=1}^{n}(\bar{h} - h_i)^2}) \times 100\% \tag{7}$$

Where $h_i^{pred}$ is the predicted bathymetry of i-th training point, $h_i$ is the measured bathymetry of i-th training point, $\bar{h}$ is the average value of the measured bathymetry of training point, and $n$ is the number of training point. $R^2$ is generally used to indicate the accuracy of training, and the greater it is, the better it is."

13. Line 261, "The SDUST2023BCO model demonstrates superior precision compared to the GEBCO_2023 and topo_25.1 models, achieving higher accuracy in 112 and 134 sub-regions, respectively, which corresponds to approximately 80.00% and 95.71% of the total sub-regions." Since the topo_25.1 model was used as input data in this study, why is the accuracy in 10 sub-regions lower than that of the topo_25.1 model? Please analyze and explain this inconsistency.
**Response:** Thanks for your suggestions and comments. Your opinions greatly helping me improve my article.

We believe that this result is reasonable, and the main reasons are:

1. When using the MLP neural network to construct seafloor topography model, the number and distribution of shipborne single-beam bathymetric data, as well as the degree of seafloor undulation, have a certain impact on the accuracy of the SDUST2023BCO model. When the number of shipborne single-beam bathymetric data is relatively small, the accuracy of the SDUST2023BCO model is lower, so we believe that the slightly lower accuracy compared to the topo_25.1 model is a normal phenomenon. For example, in the LON7_LAT1 area, the number of shipborne single-beam bathymetric data in this region is small and unevenly distributed, which has resulted in the slightly lower accuracy of the SDUST2023BCO model compared to the topo_25.1 model in this sub-region.

2. When using the MLP neural network to construct seafloor topography models, the accuracy of the SDUST2023BCO model exhibits considerable randomness, making it difficult to ensure the accuracy of bathymetric model in every sub-region. In most sub-regions discussed in this paper, the SDUST2023BCO model is more consistent with the shipborne single-beam bathymetric data, which validates the reliability of the MLP neural network and the effectiveness of the SDUST2023BCO model.

14. All these verifications show that SDUST2023BCO reaches an international advanced level of global bathymetric models. This is not an objective expression and lacks supporting evidence. Consider revising.
**Response:** Thanks for your suggestions and comments. Your opinions greatly helping me improve my article.

We revised this sentence, and the revised sentence is as follows:

"The results presented in this letter demonstrate that SDUST2023BCO reaches an international advanced level of global bathymetric models."

15. L350, the accuracy of SDUST2023BCO model is better than that of GEBCO_2023 and topo_25.1 models, especially in deeper water regions. This conclusion is not convincing. Since the shipborne single-beam bathymetry dataset was used to train your model, using the same dataset for evaluation introduces bias and compromises the validity of the conclusion.

**Response:** Thanks for your suggestions and comments. Your opinions greatly helping me improve my article.

We compare the SDUST2023BCO model, the GEBCO_2023 model, and the topo_25.1 model against shipborne single-beam bathymetric data to validate the resemblance between different bathymetric models and the shipborne single-beam bathymetric points.

We revised this sentence, and the revised sentence is as follows:

"In waters shallower than 1000m, the GEBCO_2023 model shows closer proximity to the shipborne bathymetric points compared to the topo_25.1 and SDUST2023BCO models. Overall, the SDUST2023BCO model exhibits commendable reliability in deeper waters."

16. L330, Table 8, This table only shows differences between two models, which does not provide sufficient information about accuracy. Please revise the table or provide additional data for accuracy assessment.

**Response:** Thanks for your suggestions and comments. Your opinions greatly helping me improve my article.

We aim to verify the consistency of the three models' accuracy across four regions. In these four sub-regions, the SDUST2023BCO model demonstrates good consistency with the other bathymetric models, indirectly highlighting the reliability and effectiveness of the SDUST2023BCO model.

**Reviewer #2:**

**Point 1:** In section 2.2, I think ship-borne depths should also be introduced.

**Response:** Thanks for your suggestions and comments. Your opinions greatly helping me improve my article.

In section 2.2, we added the description of the bathymetric information of shipborne single-beam points, and marked it in red font within the manuscript, with the modifications as follows:

"The 112319002 shipborne single-beam bathymetric points are used to train the MLP model which is employed to construct the SDUST2023BCO model. Among them, the largest shipborne single-beam bathymetric data is 10949.5m, and the average bathymetry is 2819.8m."

**Point 2:** Lines 161-162: According to this sentence, gravity gradients are not used as input. It that correct?

**Response:** Thanks for your suggestions and comments. Your opinions greatly helping me improve my article.

We are sorry that the expression of this sentence is inaccurate. Vertical gravity gradient is also used as input data to train MLP neural network. We have modified the expression and marked it in red font. The modification is as follows:

"In the traditional methods of constructing seafloor topography models, marine gravity data is typically used as the initial data. Based on the correlation between gravity data and bathymetry (Smith & Sandwell, 1994), the gravity anomaly, vertical gravity gradient, the meridional and prime components of vertical deflection are used as input data for training and prediction. Since MDT data can reflect the bathymetric information to a certain extent (Pujol et al., 2018; Mulet et al., 2021), MDT data has also been introduced."

**Point 3:** Line 177:Could you explain why MDT data is used as an input? What is its role?

**Response:** Thanks for your suggestions and comments. Your opinions greatly helping me improve my article.

We apologize for the unclear statement. The reason for introducing the MDT model in this paper is that the MDT model can provides bathymetric information to a certain extent. We have revised this section and added relevant references. The modifications are as follows:

"Since MDT data can reflect the bathymetric information to a certain extent (Pujol et al., 2018; Mulet et al., 2021), MDT data has also been introduced."

The impact of MDT data on the accuracy of seafloor topography models was studied in the Caribbean Sea (17-22°N, 271-280°W). Two seafloor topography models were constructed using this method described in this paper: Model1 and Model2. When constructing the Model1 model, MDT data was used; however, MDT data was not used in the construction of the Model2 model. The methods used to construct Model1 and Model2 are consistent with the method developed in this paper. Both Model1 and Model2 were interpolated to the shipborne single-beam bathymetric points using cubic spline interpolation, and the relevant statistical results are shown in Table 1.

**Table 1: Statistical results of the differences between Model1 and Model2 and shipborne single-beam bathymetric points (unit: m)**

| Model | Max | Min | Mean | STD | RMS |
|---|---|---|---|---|---|
| Model1 | 799.25 | -807.76 | 2.54 | 78.62 | 78.66 |
| Model2 | 809.26 | -806.11 | 3.56 | 82.51 | 82.59 |

As shown in Table 1, the accuracy of Model1, constructed using MDT data, improved by approximately 3.89m compared to the accuracy of Model2. This effectively validates that MDT data can enhance the accuracy of seafloor topography models.

In conclusion, the MDT data plays a vital role in the accuracy of the seafloor topography model. Using MDT data can effectively improve the accuracy of the SDUST2023BCO model. We have made revisions in the text and highlighted them in red within the manuscript.

**Point 4:** Line 322: "... 9.11m....". The precision is too high and a little unbelievable. I think this is because the correlation should be considered in Equation (7). Maybe you can do correlation analysis between these data. In addition, if you also consider the results evaluated by ship-borne depths data given in Table 4, the results would also change based on the similar equation as Equation (7).

**Response:** Thanks for your suggestions and comments. Your opinions greatly helping me improve my article.

We think this phenomenon is also reasonable. This phenomenon is because the

SDUST2023BCO model is based on the topo_25.1 model and topo_25.1 model is used as input data to train MLP neural network, so there will be a small error when calculating the STD of comparisons between the SDUST2023BCO model and topo_25.1 model by using the error propagation law.

This paper references the method used in Yuan et al. (2023), which employed the same approach to verify the accuracy of the constructed model. The Equation (7) is calculated on the basis of no correlation between different models. This method is used to verify the accuracy of SDUST2023BCO model from another angles in this paper. In addition, we modified the experiment and expression according to this suggestion.

We added a description to explain this phenomenon and marked it in red font, which is modified as follows: [Eq. (8) is due to the addition of a Equation.]

"Using Eq. (8) and the statistical results in Table 7, the $STD_S$, $STD_G$ and $STD_t$ can be calculated as 9.11m, 57.69m, and 40.18m, respectively. The high correlation between SDUST2023BCO model and topo_25.1 model causes the value of $STD_S$ to be small. This result indicates that the accuracy of the three models, from highest to lowest, is the SDUST2023BCO, topo_25.1, and GEBCO_2023 models. This effectively demonstrates that the SDUST2023BCO model has better reliability among the three models."

**Point 5:** Section 4.5 can be removed, and the contents can be put in another place.
**Response:** Thanks for your suggestions and comments. Your opinions greatly helping me improve my article.

We moved section 4.5 to section 5 according to the suggestion, and marked it in red font.

**Point 6:** Lines 437-440, they are indeed the same reference.
**Response:** Thanks for your suggestions and comments. Your opinions greatly helping me improve my article.

I'm sorry for this mistake. We carefully check the references, and some mistakes are revised and marked in red font within the manuscript.

**Point 7:** The main data used in this study is gravity field data. We know gravity field data have poor performance for the bathymetry inversion in shallow water region. Moreover, coastline is usually irregular. I wonder how the grid was designed for the coastal regions as well as the accuracy in these regions.
**Response:** Thanks for your suggestions and comments. Your opinions greatly helping me improve my article.

We are sorry that we didn't express it clearly in the manuscript. An MLP neural network trained by the shipborne single-beam bathymetric points farther than 6′ from the shore. At the same time, only the bathymetry farther than 6′ from the shore is predicted, and the area within 6' from the shore is filled by topo_25.1.

In section 3.2, we added the introduction of related processes:

"Due to the limitations in computational processing power and memory storage, an 8' × 8' grid centered on each interesting point is constructed by extending outward from each point,

as shown in Fig. 2. Grid points on the 8'×8' grid are marked from point 1 to point 64. To mitigate the impact of long-wavelength information in multi-source geodetic data, this paper uses the differences between the multi-source marine geodetic data at each grid point within an 8′×8′ area surrounding the interesting point and the multi-source marine geodetic data at the interesting point. These differences are used as the input data to train the MLP neural network. Due to some shipborne single-beam sounding points are close to the shore, and some grid points will be located on the land area. In order to improve the accuracy of SDUST2023BCO model, the shipborne single-beam sounding points farther than 6′ from the shore are used to train MLP neural network. At the same time, when modeling SDUST2023BCO model, the bathymetric values of the topo_25.1 model replace the areas within approximately 6′ from the shore."

We have carefully revised the expression of English and related contents of this paper. There are many changes in the article, but the structure and framework of this paper have not been changed. The relevant changes have been marked in red in the manuscript.

We appreciate for reviewer's warm work earnestly, and hope that the correction will meet with approval.

Once again, thank you very much for your comments and suggestions.

**References**

Annan, R. F. and Wan, X.: Recovering bathymetry of the Gulf of Guinea using altimetry-derived gravity field products combined via convolutional neural network, Surv. Geophys, 43, 1541-1561, https://doi.org/10.1007/s10712-022-09720-5, 2022.

Mulet, S., Rio, M., Etienne, H., Artana, C., Cancet, M., Dibarboure, G., and Feng, H. et al.: The new CNES-CLS18 global mean dynamic topography, Ocean Sci., 17, 789-808, http://doi.org/10.5194/os-17-789-2021, 2021.

Pujol, M. I., Schaeffer, P., Faugère, Y., Raynal, M., Dibarboure, G., and Picot, N.: Gauging the improvement of recent mean sea surface models: A new approach for identifying and quantifying their errors, J. Geophys. Res. Oceans, 123, 5889-5911, http://doi.org/10.1029/2017jc013503, 2018.

Yuan, J., Guo, J., Zhu, C., Li, Z., Liu, X., and Gao, J.: SDUST2020 MSS: a global 1′ × 1′ mean sea surface model determined from multi-satellite altimetry data, Earth Syst. Sci. Data, 15, 155–169, https://doi.org/10.5194/essd-15-155-2023, 2023.

Zhou, S., Liu, X., Sun, Y., Chang, X., Jia, Y., Guo, J., and Sun, H.: Predicting bathymetry using multisource differential marine geodetic data with multilayer perceptron neural network, Int. J. Digit. Earth, 17, 2393255, https://doi.org/10.1080/17538947.2024.2393255